# Validity and Reliability of Short-Term Heart Rate Variability Parameters in Older People in Response to Physical Exercise

**DOI:** 10.3390/ijerph20054456

**Published:** 2023-03-02

**Authors:** Matías Castillo-Aguilar, Matías Mabe Castro, Diego Mabe Castro, Pablo Valdés-Badilla, Tomás Herrera-Valenzuela, Eduardo Guzmán-Muñoz, Morin Lang, Oscar Niño Méndez, Cristian Núñez-Espinosa

**Affiliations:** 1Centro Asistencial de Docencia e Investigación (CADI-UMAG), University of Magallanes, Punta Arenas 6200000, Chile; 2Kinesiology Department, University of Magallanes, Punta Arenas 6200000, Chile; 3School of Medicine, University of Magallanes, Punta Arenas 6200000, Chile; 4Department of Physical Activity Sciences, Faculty of Education Sciences, Universidad Católica del Maule, Talca 3480094, Chile; 5Carrera de Entrenador Deportivo, Escuela de Educación, Universidad Viña del Mar, Viña del Mar 2520000, Chile; 6Department of Physical Activity, Sports and Health Sciences, Faculty of Medical Sciences, Universidad de Santiago de Chile (USACH), Santiago de Chile 9170022, Chile; 7Escuela de Kinesiología, Facultad de Salud, Universidad Santo Tomás, Talca 3480094, Chile; 8Department of Rehabilitation Sciences and Human Movement, Faculty of Health Sciences, Universidad de Antofagasta, Antofagasta 1270300, Chile; 9Center for Research in Physiology and Medicine of Altitude, Biomedical Department, Faculty of Health Sciences, Universidad de Antofagasta, Antofagasta 1270300, Chile; 10Facultad de Ciencias del Deporte y la Educación Física, Universidad de Cundinamarca, Bogotá 252211, Colombia; 11Interuniversity Center for Healthy Aging, Chile 3480094, Chile

**Keywords:** older adults, physical activity, heart rate, autonomic nervous system

## Abstract

Background: Currently, and to the best of our knowledge, there is no standardized protocol to measure the effect of low- to moderate-intensity physical exercise on autonomic modulation focused in older people. Aim: Validate a test–retest short-term exercise protocol for measuring the autonomic response through HRV in older people. Methods: A test–retest study design was used. The participants were selected through intentional non-probabilistic sampling. A total of 105 older people (male: 21.9%; female: 78.1%) were recruited from a local community. The assessment protocol evaluated HRV before and immediately after the 2-min step test. It was performed twice on the same day, considering a time of three chronological hours between the two measurements. Results: The posterior distribution of estimated responses in the Bayesian framework suggests moderate to strong evidence favoring a null effect between measurements. In addition, there was moderate to robust agreement between heart rate variability (HRV) indices and assessments, except for low frequency and very low frequency, which showed weak agreement. Conclusions: Our results provide moderate to strong evidence for using HRV to measure cardiac autonomic response to moderate exercise, suggesting that it is sufficiently reliable to show similar results to those shown in this test–retest protocol.

## 1. Introduction

The autonomic nervous system (ANS) has a role in the modulation of a plethora of physiological processes, whereas the balance between the sympathetic (SNS) and parasympathetic (PNS) branches of the ANS has the potential to influence the cardiovascular response to some forms of physical stress, such as those imposed by the environment in the form of exercise [1], and those derived from internal physiological processes such as aging [2]. Although both the PNS and SNS constantly interact to maintain autonomic balance, current evidence indicates that the SNS continuously influences the electrophysiological properties of the heart, while the PNS would have a sympathetic-braking effect that could modulate cardiac activity [3].

Understanding how the ANS communicates with the heart has potential implications for clinicians searching for new therapeutic strategies, especially those aimed at creating novel monitoring tools and cost-effective biomarkers for the aging population [2].

The cardiac modulatory effects of ANS can be examined noninvasively through changes in the time intervals between successive R-R intervals in the heart, also known as heart rate variability (HRV) [4]. Moreover, there has been a growing interest in evaluating the HRV response to physical exercise since it reflects a functional autonomic modulation in relation to the active life of people from health to disease [1,5,6,7,8,9,10,11,12,13]. Evidence shows that effort-related cardiovascular reactivity is associated with executive function and physical fitness [14,15]. Moreover, evidence suggests a sex-dependent response in younger individuals to physical stress [16].

This is particularly interesting in older people, given that during the aging process, multiple physiological, cognitive, and social aspects influence cardiovascular parameters associated with health and functional outcomes in this population [17,18,19]. Thus, environmentally mediated alterations in ANS may derive from pathological states frequently associated with adverse clinical effects that can be monitored or even predicted in relation to the possible alterations that the subject can demonstrate in the autonomic register after constant and individualized monitoring [20,21]. Changes in cardiac autonomic modulation occur during aging, resulting in reduced vagal tone and increased sympathetic activity predominance [22].

In a recent study, a group of frail and non-frail older people was compared, assessing the HRV response to a 160-m walking test, showing that frail older people exhibit an impaired response compared to non-frail individuals [23]. Although similar evidence exists in this regard [24,25,26], it is already known that the differences in the autonomic modulation of individuals can be influenced by the type, duration, and intensity at which the exercise is performed [27].

To the best of our knowledge, no standardized protocols measure the effect of low- to moderate-intensity physical exercise on autonomic modulation focused in older adults. For this reason, this study aims to validate a test–retest short-term exercise protocol for measuring the autonomic response through HRV in older people. We hypothesized that validating a test–retest short-term exercise protocol would yield good reproducibility of cardiac autonomic activity in older adults.

## 2. Materials and Methods

### 2.1. Study Design

A test–retest study design was used. The study aims and assessment procedures were explained to all the subjects. The participants were selected through intentional non-probabilistic sampling, distributed between females and males. The participants were monitored for two consecutive moments on the same day. The first measurement was obtained in the morning between 09:00 and 10:00 h and the second measurement, three hours after the previous measurement was completed. During the three hours of rest, the participants completed surveys (quality of life and personal history) and rested mentally and physically within the same evaluation building. It is also important to note that the testing area was at an average room temperature of 21 °C and had comfortable chairs for each of the participants to rest.

### 2.2. Participants

Participants were recruited from a local community. Subjects were included if (i) they were aged 60 years or older; (ii) were permanently residing in the Magallanes and Chilean Antarctic region; (iii) had a percentage greater than 60% on the Karnofsky Performance Status Scale, which allowed us to work with older people who had a state of autonomy necessary to carry out the study tests; (iv) absence of the following diagnosis: diabetic neuropathy; use of pacemakers; clinical depression; cognitive or motor disability; and dementia. The exclusion criteria were: (i) consumption of beta-blockers during the study, (ii) taking drugs or stimulant substances within 12 h before the cardiac assessment; and (iii) having some degree of motor disability that prevented participants from moving around. No participants met the exclusion criteria. The volunteers were informed about the study’s aims, procedures, responsibilities, and risks. All participating subjects gave their permission and provided informed consent before participation. The Ethics Committee of the University of Chile (ACTA N°029-18/05/2022) and the Ethics Committee of the University of Magallanes (N°008/SH/2022) approved this study following the regulations established by the Declaration of Helsinki on ethical principles in human beings.

### 2.3. Procedures

The measured protocol consisted of the evaluation of HRV before and immediately after the 2-min step test, which is a part of the Senior Fitness Test protocol [28]. It consisted of a functional cardiorespiratory test, where each subject marched on the site as many times as possible for 2 min. The protocol was performed twice on the same day, considering a time of three chronological hours between the two measurements. This time has been deemed sufficient for cardiovascular recovery for the level of intensity of the exercise performed [27,29]. During the waiting time between each execution period, the participants rested and performed daily activities such as walking, going to the bathroom, and resting comfortably in chairs, to recover their basal states before the execution of the next protocol. Throughout the assessment, the participants were monitored using cardiovascular measures (i.e., heart rate and blood pressure) to monitor the absence of adverse events while applying the exercise protocol. The evaluation protocol was estimated to last approximately 20 min for each subject assessed at each time. None of the participants expressed discomfort during the evaluation.

### 2.4. Assessments

#### 2.4.1. Morphological Measures

Body mass (kg) and total body fat (%) were assessed by bioimpedance using the Tanita BC-558 Ironman Segmental Body Composition Monitor (Tanita Ironman, Arlington Heights, IL 60005, USA), with a concordance of 89.3% compared to the Dual X-ray Absorption test using standard measurement protocols [30,31]. Height was measured by a CHARDER^®^ HM230M manual height rod (Charder Electronics Co., Ltd., No. 103, Guozhong Rd., Taichung City, Taiwan).

#### 2.4.2. Cardiovascular Parameters

Systolic blood pressure (SP) and diastolic blood pressure (DP) were measured (Omron^®^ Pressure Monitor). As part of the protocol, we ensured that the participant had an SP less than 140 mmHg and a DP less than 90 mmHg to start the HRV measurements. Cardiac autonomic modulation was determined via a recording of RR intervals obtained by the Polar Team2 system (Polar^®^) application. The volunteers remained seated in a chair during the entire HRV measurement procedure, and RR intervals were recorded continuously during the last 10 min of rest, and were subsequently analyzed for 5 min. The subject’s breathing rate was spontaneous. Artifacts and ectopic heartbeats (which did not exceed 3% of the recorded data) were excluded [32]. The time-domain parameters considered for the analysis were the square root of the mean squared differences of the successive RR intervals (RMSSD, expressed in ms) as an index of parasympathetic activity [33] and the standard deviation of the RR intervals (SDNN), which reflects the total variability, that is, the sympathetic and parasympathetic contribution of the ANS on the heart [34,35]. In the frequency domains, the high-frequency (HF) power band was considered, given that it reflects the parasympathetic influences on heart rate (HR) and includes respiratory sinus arrhythmia [36], as well as the low-frequency (LF) band, given that it is associated with baroreflex activity [37]. Moreover, the very-low-frequency band (VLF) was considered, due to its strong association with emotional stress [19,38].

Additionally, the Stress Index (SI) and Parasympathetic and Sympathetic Nervous System Index (PNS and SNS, respectively) were calculated. The PNS Index, reflecting total vagal stimulation, was calculated from the mean R-R intervals, RMSSD, and Poincaré Plot Index SD1 in normalized units (linked to RMSSD) and reflects how many standard deviations above or below the normal population averages the obtained values were. The SNS Index, reflecting total sympathetic stimulation, was calculated from mean R-R intervals, Baevsky’s Stress Index (a positively related value to cardiovascular system stress and cardiac sympathetic activity), and the Poincaré Plot Index SD2 in normalized units (related to SDNN) and its interpretation was similar to the PNS Index [34,39]. The SI is an indicator that represents the degree of load on the ANS control system [40]. It is normalized by using the square root of Baevsky’s SI [41]. All the analyses were performed with the software Kubios HRV^®^ (Kuopio, Finland).

To ensure the relative intensity achieved during the test, the median percentage of maximum HR (%HRmax) performed during the TMST was used as a measure of relative intensity, which was reported for each stage of the TMST measurements.

##### 2.4.3. 2-Minute Step Test

The 2-min step test (TMST), a sub-test of the Senior Fitness Test protocol [42], was performed. The test consisted of a two-minute step test to assess cardiorespiratory fitness, recording the number of knee raises that reached at least a 70° angle on the thigh-femoral joint of each participant. Vital signs were evaluated through blood pressure, and the participant’s well-being was visually checked, prioritizing that they breathed normally, wore comfortable clothes, and felt physically and mentally fit to perform the test. Then the participants stood next to a wall and had their iliac crest and patella measured, both measurements were marked on the wall. Subsequently, a mark was assigned in the middle of these two distances, indicating the point where they had to raise the knee during the execution of the test. At the signal “go”, the participant began walking (not running) on the spot, raising each knee to the mark on the wall as many times as possible in the 2 min. Only the number of times the right knee reached the required height was counted. That was the score. After the test, the older person walked slowly for one minute to cool down. Throughout the test, the participant was monitored through HRV.

### 2.5. Statistical Analysis

We fitted a constant (intercept-only) Bayesian linear model, estimated using Markov chain Monte Carlo sampling (MCMC algorithm) with five chains of 50,000 iterations and a warm-up of 25,000, to describe the variations between analogous assessments of HRV between the two measurements of the TMST (e.g., variations between the previous measurement from the first and second TMST assessment, i.e., both pre-TMST measurements). A Gaussian distribution centered around zero (μ=0) with the scale specified as 2.5 times the standard deviation (SD) of the response measurement (σ=2.5×sd(y)), was used as priors, given that we were assuming that there was no variation between analogous assessments in between the two TMST measurements, thus μ=0 reflecting that expectation. The prior’s scale for each model can be seen in Table A1 in Appendix A. For each model, extreme outliers were removed using box plot methods, considering values above Q3+(3×IQR) or below Q1−(3×IQR).

In addition to previous models, Lin’s concordance correlation coefficient (CCC) and its 95% credible interval (CI95%) were computed as a measure of agreement between analogous HRV assessments through the Bayesian approach for the multivariate normal [43] and labeled according to Evans [44] as follow: CCC < 0.2, Very weak; 0.2 ≤ CCC < 0.4, Weak; 0.4 ≤ CCC < 0.6, Moderate; 0.6 ≤ CCC < 0.8, Strong; CCC ≥ 0.8, Very strong.

The use of the Bayesian framework represents an advantage over the classical frequentist paradigm since it allows a direct interpretation of the results, given that full access to the posterior distribution can be obtained, which allows a better understanding of the associated uncertainty of the model parameters [45,46].

Following the Sequential Effect eXistence and sIgnificance Testing (SEXIT) framework to describe the effects from Bayesian models [47], the median and the CI95% (using the highest density interval) were reported as a measure of centrality and uncertainty, the probability of direction (pd) as the measure of existence, the proportion of the posterior probability distribution that falls outside the region of practical equivalence (ROPE) as a measure of practical significance, estimated as one-tenth (1/10 = 0.1) of the SD of the response variable, and Bayes factor (BF10) using Savage–Dickey density ratio against the point null indicating if the null value has become less or more likely given the observed data [48], using this as a measure of an absolute magnitude of evidence in favor or against the null hypothesis (of no effect).

For BF interpretation, we considered: BF = 1, no evidence; 1 < BF ≤ 3, anecdotal; 3 < BF ≤ 10, moderate; 10 < BF ≤ 30, strong; 30 < BF ≤ 100, very strong; and BF > 100, as extreme evidence [49]. For the proportion of the posterior in the ROPE we considered: <1%, significant; <2.5%, probably significant; ≤97.5% and ≥2.5%, undecided significance; >97.5%, probably negligible; >99%, negligible [47]. The convergence and stability of Bayesian sampling have been assessed using R-hat, which should be below 1.01 [50], and the Effective Sample Size (ESS), which should be greater than 1000 [51].

All computations were performed using the R programming language for statistical computing on version 4.2.1 [52], and complementary R packages [53,54,55,56].

## 3. Results

A total of 105 participants (male, *n* = 23 [21.9%]; female, *n* = 82 [78.1%]) were enrolled in the study. Sample characteristics and body composition parameters can be observed in Table 1.

When analyzing the variations within TMST measurements, we observed strong evidence in favor of the null hypothesis on post-TMST for all domains except for stress index, whereas the evidence in favor of the null was moderate. At pre-TMST, we found strong evidence in favor of the null for RMSSD, HF, and the SNS Index; moderate evidence in favor of the null for VLF; and anecdotal evidence in favor of the null for LF. Despite the previous findings, we found very strong evidence in favor of the null hypothesis for PNS, suggesting a significant shift from pre-TMST-1 to pre-TMST-2; moderate evidence against the null for SDNN, mean R-R, and anecdotal evidence against the null for the stress index. The posterior distribution of each Δ associated with TMST measures and the trace plots indicating the convergence of each model can be seen in Figure 1 and Figure 2, respectively. The summary statistics of the posterior distribution of each model can be seen in Table 2.

During the assessments, we observed that the %HRmax achieved during the measurements at TMST-1 was 66.7% CI_95%_[45.9%, 85.1%] and at TMST-2, the median %HRmax was 68.1% CI_95%_[46.5%, 88.3%] during measurements. The recorded %HRmax throughout the TMST assessments can be seen in Figure 3.

The concordance and agreement limits between measurements can be seen in Table 3 and Figure 4, respectively.

## 4. Discussion

This study aimed to validate a test–retest short-term exercise protocol for measuring the autonomic response through HRV in older people to physical exercise. As HRV is highly sensitive to internal and external factors such as sleep quality and mood [57,58], we conducted a test–retest protocol repeated during the same day to minimize the influence of factors other than exercise between assessments.

To the best of our knowledge, this is the first study on the use of HRV to measure the cardiac autonomic response to exercise in older adults. This is a significant advancement, given that HRV is a valuable tool for identifying patients at cardiovascular risk and predicting all-cause mortality, neurological disorders, and worse quality of life in older adults [59,60,61].

The assessment of autonomic response to functional physical activity has enormous potential in the clinical setting. It allows for standardized assessments that provide insights into physiological adaptations to activities in functional contexts and overall health status in older adults [62,63]. The selection of a low to moderate aerobic exercise modality is particularly noteworthy as it represents the well-functional spectrum of activities of daily living during aging. Hence, it is a valuable tool for identifying early indications of cardiovascular disease and other age-related health issues.

This study provides robust evidence that short-term HRV measurement is reliable and valid for assessing the cardiac autonomic response to moderate exercise in older adults. The consistency and reproducibility of HRV measurements in both assessments demonstrated that they accurately captured the autonomic response to the exercise protocol, thus supporting the method’s reliability in this population. These findings align with previous research that has demonstrated the efficacy of HRV as a tool for evaluating autonomic function in various populations, including older adults [1,5,6,7,8,9,10,11,12,13]. Using HRV measurements for this purpose could help clinicians identify and monitor individuals at higher risk, leading to earlier interventions and better outcomes. Moreover, this study used Bayesian data analysis to strengthen its statistical methods. The Bayesian analysis allows for prior knowledge and uncertainty to be integrated into the analysis, leading to more precise estimates and efficient use of data. This approach was used to estimate the reliability and validity of the HRV measurements, resulting in a more accurate and informative assessment of the autonomic response to exercise in older adults. This valuable contribution to the field of exercise science could encourage other researchers to adopt this approach in their research.

However, this study is not without limitations. We included individuals who suffered from different non-cardiac diseases, which could have affected the HRV measurements. Additionally, the use of non-probabilistic sampling may have affected the generalizability of our results. To address these limitations, future research could use more specific eligibility criteria to minimize the influence of factors other than exercise on the autonomic response, and a longitudinal design could be used to examine the effects of exercise on HRV over time.

Overall, the findings of this study have important implications for the use of HRV in assessing the cardiac autonomic response to exercise in older adults. By validating a test–retest short-term exercise protocol, this study provided a reliable and standardized method for measuring physiological adaptations to activities in functional contexts and health status in older adults. This, in turn, could lead to earlier identification of patients at cardiovascular risk and the development of more targeted interventions that improve health outcomes in this population. The use of HRV in future research and clinical practice could significantly impact the health and well-being of older adults. This research has significant implications for advancing the knowledge of functional autonomic assessments during physical exercise and providing a foundation for further research in this area.

## 5. Conclusions

Our results provided moderate to strong evidence for using HRV to measure cardiac autonomic response to moderate exercise, suggesting that it is sufficiently reliable to show similar results to those shown in this test–retest protocol. These findings have significant implications for clinicians involved in active aging programs, as HRV is a valuable clinical tool that can be used to identify patients at cardiovascular risk and tailor interventions accordingly.

## Figures and Tables

**Figure 1 ijerph-20-04456-f001:**
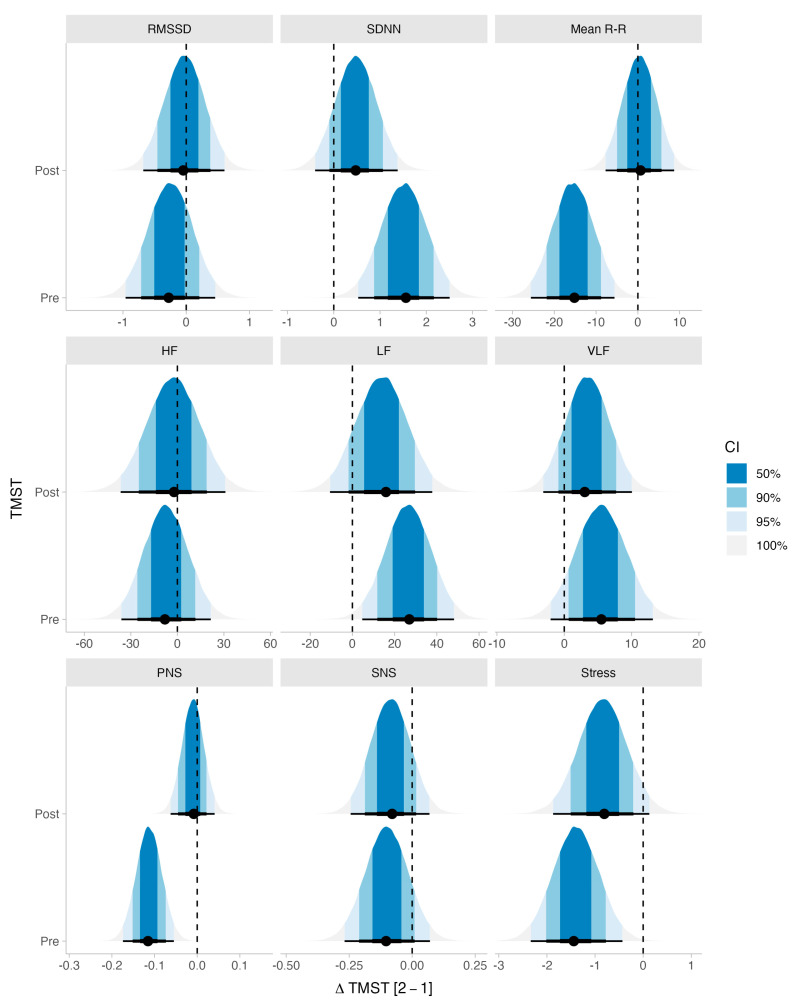
Posterior distributions for each HRV measure for each difference between 2-min step test pre- and post-assessments (∆HRV-TMST [2-1]). This represents the uncertainty around each estimate and the probability associated with it.

**Figure 2 ijerph-20-04456-f002:**
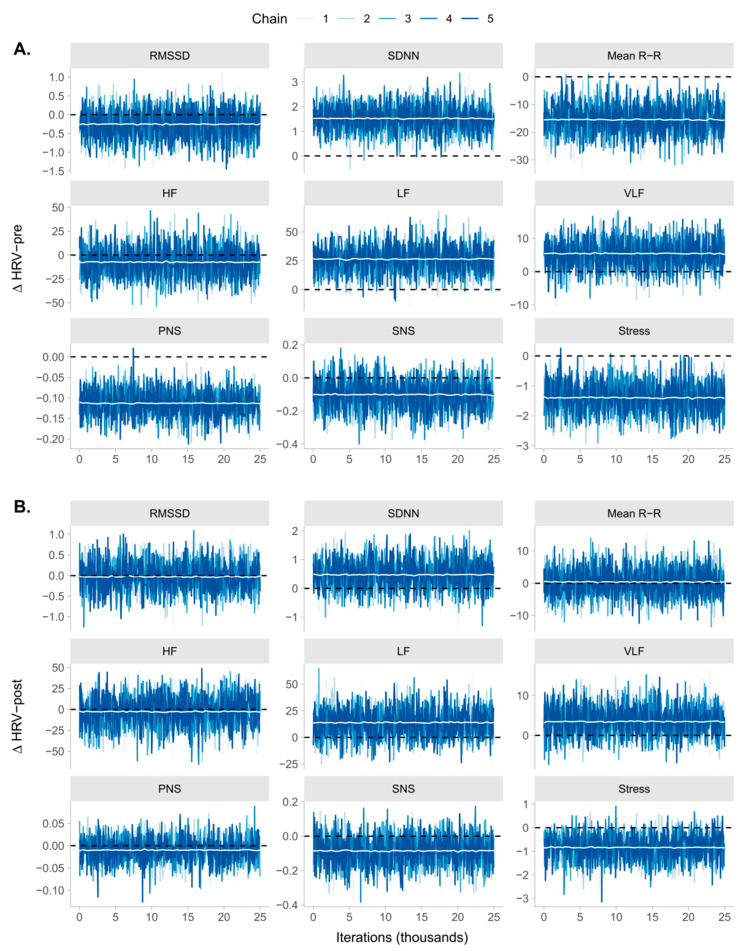
Trace plots indicating the estimated values for ∆HRV-TMST [2-1] on each iteration (horizontal axis) for each MCMC chain. The white line at the center of each trace plot represents the median value from all chains in 500-iteration intervals. (**A**) Model diagnostics for HRV parameters pre-TMST; (**B**) model diagnostics for HRV parameters post-TMST.

**Figure 3 ijerph-20-04456-f003:**
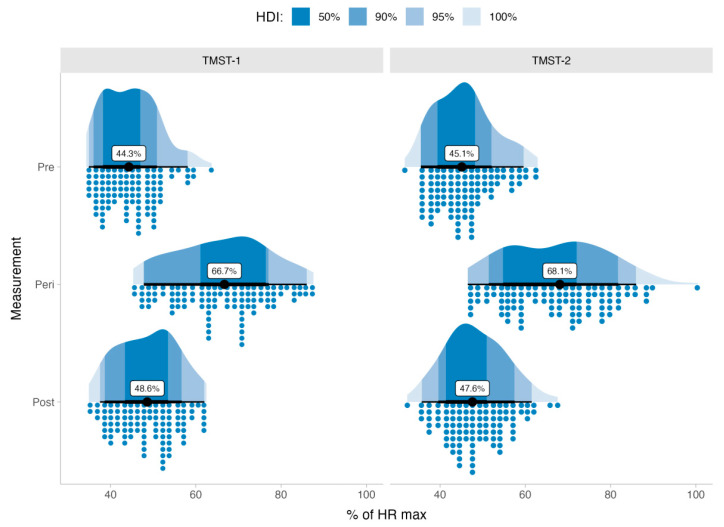
Raincloud plots of the mean heart rate for each measurement during both 2-min step test protocols. The black line at the center of each Raincloud plot represents the median value observed from the data.

**Figure 4 ijerph-20-04456-f004:**
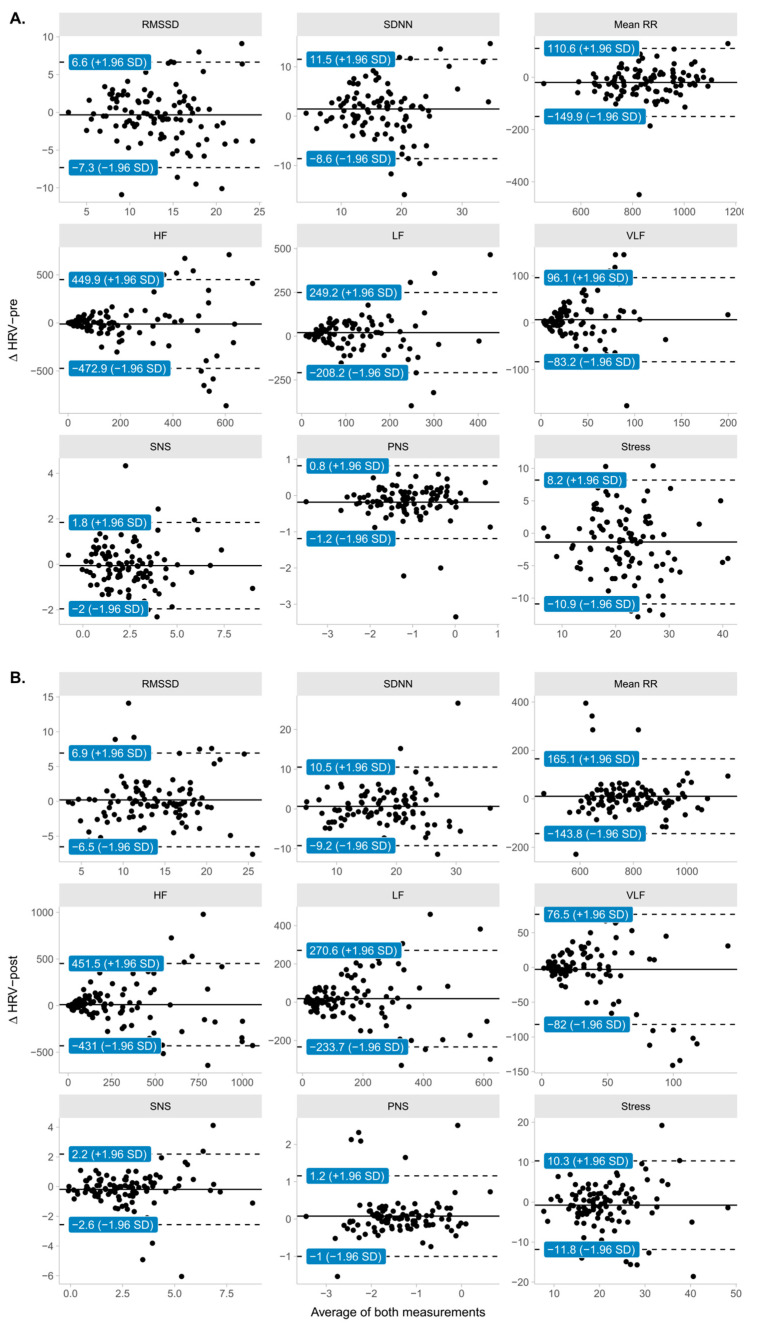
Bland–Altman plots and 95% limits of agreement between pairs of measurements in correlative assessments relative to the application of the TMST. (**A**) Limits of agreement for HRV parameters pre-TMST; (**B**) Limits of agreement for HRV parameters post-TMST.

**Table 1 ijerph-20-04456-t001:** Overall and aggregated by sex descriptive statistics of body composition parameters from the study sample.

Characteristic	Overall, N = 105 ^1^	Sex		Comparison	
		Female, N = 82 ^1^	Male, N = 23 ^1^	Difference ^2^	95% CI ^2,3^
Age (years)	70.9 ± 5.9	70.3 ± 6.0	73.2 ± 5.0	−0.54	−1.0, −0.07
Body mass (kg)	74 ± 14	74 ± 15	77 ± 10	−0.24	−0.71, 0.22
Height (cm)	155 ± 8	153 ± 6	164 ± 7	−1.7	−2.2, −1.1
Body fat (%)	38 ± 9	41 ± 6	25 ± 6	2.8	2.2, 3.4
Body water (%)	47 ± 6	45 ± 4	55 ± 5	−2.3	−2.9, −1.8
Bone mass (%)	2.74 ± 4.17	2.70 ± 4.73	2.87 ± 0.28	−0.05	−0.51, 0.41
Muscle mass (%)	44 ± 8	41 ± 5	54 ± 6	−2.5	−3.1, −1.9

^1^ Mean ± SD, ^2^ Standardized Mean Difference, ^3^ CI = Confidence Interval.

**Table 2 ijerph-20-04456-t002:** Summary and model diagnostics for each HRV measurement within TMST assessments. CI, credible interval; pd, probability of direction; ROPE, region of practical equivalence; ESS, effective sample size; BF, Bayes factor.

Measure	Median	CI 95% ^1^		pd ^3^	ROPE ^2^			R-hat ^4^	ESS ^5^	BF ^6^
		CI Low	CI High		Low	High	% Inside			
∆HRV-pre										
RMSSD	−0.25	−0.959	0.458	75.8%	−0.365	0.365	58%	1.000	83,974	0.050
SDNN	1.52	0.529	2.506	99.8%	−0.511	0.511	2.3%	1.000	84,522	3.617
Mean R-R	−15.47	−25.602	−5.593	99.8%	−5.151	5.151	2.1%	1.000	84,462	3.838
HF	−7.38	−36.091	21.513	69.3%	−13.735	13.735	59.6%	1.000	83,517	0.048
LF	26.44	4.661	48.151	99.1%	−10.659	10.659	7.6%	1.000	80,397	0.720
VLF	5.45	−2.030	13.162	92.2%	−3.742	3.742	31.9%	1.000	80,128	0.112
PNS	−0.11	−0.174	−0.055	100%	−0.030	0.030	0.3%	1.000	84,078	32.902
SNS	−0.10	−0.268	0.071	88%	−0.087	0.087	42.1%	1.000	82,412	0.079
Stress	−1.40	−2.336	−0.430	99.8%	−0.487	0.487	3.1%	1.000	82,470	2.526
∆HRV-post										
RMSSD	−0.03	−0.679	0.605	53.3%	−0.330	0.330	68.5%	1.000	85,628	0.040
SDNN	0.48	−0.403	1.382	85.5%	−0.452	0.452	45.5%	1.000	84,183	0.070
Mean R-R	0.41	−7.730	8.686	53.9%	−4.139	4.139	67.5%	1.000	85,178	0.040
HF	−2.62	−36.558	30.969	56.2%	−16.815	16.815	66.8%	1.000	83,878	0.041
LF	13.92	−10.604	37.828	87%	−12.124	12.124	42.5%	1.000	81,263	0.077
VLF	3.46	−3.116	10.053	85.1%	−3.253	3.253	45.4%	1.000	85,824	0.070
PNS	−0.01	−0.062	0.041	65.1%	−0.026	0.026	64.3%	1.000	82,840	0.044
SNS	−0.09	−0.244	0.069	86.1%	−0.079	0.079	44.4%	1.000	80,284	0.072
Stress	−0.84	−1.871	0.128	95.1%	−0.505	0.505	24.9%	1.000	81,326	0.158

^1^ CI = Credible interval; ^2^ ROPE = Region of practical equivalence; ^3^ pd = Probability of direction; ^4^ R-hat = Potential scale reduction factor; ^5^ ESS = Effective sample size; ^6^ BF = Bayes factor.

**Table 3 ijerph-20-04456-t003:** Lin’s concordance correlation coefficients evaluate the agreement between analogous HRV measurements.

Parameter	CCC ^2^	Interpretation ^3^	CI 95% ^1^	
			Low	High
∆HRV-pre				
RMSSD	0.722	Strong	0.624	0.808
SDNN	0.651	Strong	0.534	0.751
Mean RR	0.866	Very Strong	0.817	0.911
HF	0.449	Moderate	0.285	0.601
LF	0.376	Weak	0.201	0.540
VLF	0.335	Weak	0.159	0.502
SNS	0.839	Very Strong	0.778	0.893
PNS	0.769	Strong	0.687	0.840
Stress	0.716	Strong	0.614	0.800
∆HRV-post				
RMSSD	0.755	Strong	0.666	0.832
SDNN	0.712	Strong	0.608	0.801
Mean RR	0.824	Very Strong	0.758	0.881
HF	0.703	Strong	0.599	0.794
LF	0.656	Strong	0.538	0.761
VLF	0.361	Weak	0.199	0.520
SNS	0.780	Strong	0.698	0.849
PNS	0.755	Strong	0.669	0.832
Stress	0.726	Strong	0.628	0.811

^1^ CI = Credible Interval; ^2^ CCC = Lin’s concordance correlation coefficient; ^3^ Interpretation according to Evans (1996) [44].

## Data Availability

The raw data supporting the conclusions of this article are available from the authors without undue reservation.

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
