# Peer review of "Validity and Reliability of Short-Term Heart Rate Variability Parameters in Older People in Response to Physical Exercise"

_ijerph, 2023, doi:10.3390/ijerph20054456_

Round 1
Reviewer 1 Report
Congratulations. The subject of the study is very interesting, however, the presentation of the manuscript must be improved. Please, find my comments in the attached file.

Author Response
Dear reviewer, we greatly appreciate your feedback. We have responded to each comment made detailing your comment and this response. Also, the new manuscript has an option for change so that you can observe the improvements in our work.
|
Comment |
Response |
|
Authors must include a clear aim of the study. |
Aim added to the abstract. |
|
Authors must define all the abbreviations for the first time in the abstract and throughout the manuscript. |
All abbreviations were define throughout the manuscript as suggested. |
|
- Correct...Low Frecuency and Very Low Frecuency...to....low frequency and very low frequency - Add some information in the Methods about ...low frequency and very low frecuency |
Frecuency changed to frequency as suggested. |
|
(line: 40-44) This sentence is too long. I suggest dividing it in two or three senteces. |
The sentence was divided in two sentences as suggested. |
|
(line: 55) It is non sense...I suggest deleting "however". I suggest using ...moreover.. |
The word “However” was changed to “Moreover” as suggested. |
|
(line: 57; 74 and References) I suggest adding more references here. |
References were added as suggested. |
|
(line: 87) rewrite...09:00 and 10:00...to...09:00 and 10:00 h |
Modified as suggested. |
|
(line: 92) Move to the Results section |
Modified as suggested. |
|
(line: 109) I suggest describing briefly the protocol. |
The protocol has already been briefly described as suggested in lines 110-112, however, minor spell issues were addressed. |
|
(line: 147) Add reference...all the abbrevitions must be define in the first time in the manuscript |
Abbreviations were added according to the order in which they appear in the manuscript. |
|
(line: 163) Clarify |
Blood pressure was checked, and people felt well before the test. The visual check consists of seeing if the subject breathes well, if they feel comfortable in the place of execution of the test and if they have any motor, physical or mental difficulties before performing the test. |
|
(line: 165) clarify, and if possible, use a Figure |
The text was modified to give a better explanation of the protocol carried out. |
|
(line: 203) please, follow the title of the Table 1. |
We do not understand the reviewer's suggestion, however the paragraph and table were modified |
|
(Table 1) Add the unity of each parameter: age, body mass, height,...... |
Units were added as suggested. |
|
(Table 1) I suggest using "body mass" instead of weight |
Modified as suggested. |
|
(line: 227) clarify and add information in the "Methods section". |
The rationale behind using deltas is to describe variations between analogous measurements of HRV, as described in lines 174-177 from the statistical analyses section. |
|
(line: 233; 247 and Table 3) Clarify in the "Methods" section. |
- The rationale behind deltas is to describe variations between analogous measurements of HRV, as described in lines 174-177. - %HRMax was added in the section 2.4.2 explaining the rationale of this metric. - Lin's concordance correlation coefficient, as well as Evans conventions for their interpretation are described in lines 184-188. |
|
(section: Discussion) The Discussion section is too short. I suggest improving it. |
The discussion section was modified accordingly to suggestions. |
|
(line: 282) Clarify "these". |
The discussion section was modified accordingly to suggestions |
|
(section: Discussion) I suggest adding a paragraph before the Conclusion with the strenght of the study. |
The discussion section was modified accordingly to suggestions |
|
(section: References) Some references are too old. If possible, I suggest replacing them. |
We have improved the references without removing the old ones that seem relevant to the field of study in which this study is located. |
|
(line: 415 and 420) I suggest adding a link |
Link added as suggested |
Reviewer 2 Report
The manuscript adds to the limited body of knowledge on exercise and autonomic modulation and could be important with regard to the use of HRV as a measure of the autonomic response. It also adds to the current knowledge regarding age and geographic location.
The introduction clearly describes the background and need for the study, although some sentences could have been more specific (such as lines 61-68, and 71-74. There are some grammar errors, such "this study aim validate" -> this study aims to validate. This should be checked throughout the manuscript.
Line 83: I am not sure that a test-retest should be defined as a cross-sectional study, although the time between tests may be short. The number of subjects are adequate for a validation study.
Line 114-115: Older people could perhaps be replaced with The participants.
Lines 142-146 are unclear and needs improvement.
Author Response
Dear reviewer, we appreciate your feedback. We have responded to each comment made detailing your comment and this response. Also, in the new manuscript is the option of change so that you can observe the improvements in our work.
|
Comment |
Response |
|
(line: 83) I am not sure that a test-retest should be defined as a cross-sectional study, although the time between tests may be short. The number of subjects are adequate for a validation study. |
We had described the work as a Cross-Sectional Study, due to the time in which the retest was carried out, however, we decided to describe it only as a Test-Retest study. We appreciate the suggestion |
|
(line: 114-115) Older people could perhaps be replaced with The participants. |
“Older people” was modified to “the participants” in the results and methodology but not in the introduction and discussion, as it reflects the intended target population. |
|
Lines 142-146 are unclear and needs improvement. |
The phrasing was improved according to suggestions. |
|
There are some grammar errors, such "this study aim validate" -> this study aims to validate. |
This was corrected as suggested. |
|
The introduction clearly describes the background and need for the study, although some sentences could have been more specific (such as lines 61-68, and 71-74) |
The introduction was modified to further specify the importance of the topic of the study |
Reviewer 3 Report
An interesting study with a good presentation of the information. However, there are some amendments and further clarification is needed in some areas of the manuscript.
Most of the comments are provided as an attachment. and some are mentioned here. The discussion part is very short, there was no confirmation whether the study met the hypothesis or not.
the aim of the study is not clear, and the justification or rationale is not convincing. Therefore, I suggest to be improved,

Author Response
Dear reviewer, we appreciate your feedback. We have responded to each comment made detailing your comment and this response. Also, the new manuscript is with the option of change, so that you can observe the improvements in our work.
|
Comment |
Response |
|
(line: 65-66) Need Further explanation, how Alteration in ANS can be predicted? |
We have modified the sentence to give a better explanation. |
|
(line: 76) What intensity of physical exercise? |
Intensity of exercise was added as suggested. |
|
(line: 86) What sort of monitoring? |
The monitoring procedure was further described in the Procedures section |
|
(line: 89) What kind of surveys? |
This moment was used so that the elderly could answer quality of life and personal history surveys, which were not extensive and did not require much attention from the participants, favoring the greatest possible rest before the retest. This point is clarified in the text. |
|
(line: 89-90) More information is needed, like how cool or warm the area of testing, whether the participants were seated comfortably during the testing intereval. |
We have clarified this point in the text, indicating that the testing room was at 21°C and that it did have comfortable chairs for the participants to rest. Thanks a lot for the suggestion |
|
(line: 95) Justify the use of this scale? |
We have added a sentence with the justification in the text. We chose this scale because it was designed to measure the level of activity of the patient and generally assess the autonomy of the subject, with which we can ensure that the participant could carry out the activities relevant to this study. |
|
(line: 115) Give examples? |
We have explained this sentence better, indicating that daily activities and rest are the ones that the subjects mainly carry out, with the aim of recovering their basal state before the next protocol. |
|
(line: 137-139) The time domain is mainly chosen during longer measurements 24hr, wherase, short period 5min the frequency domain is much preferred? Justify why time doian was choosen? |
Both frequency and time domains have been used in short periods of time to determine the autonomic response of a subject. Frequency domains can be largely influenced by the systemic responses of the individual, while frequency domains are considered as less influenced by these parameters and better interpret the autonomic response in shorter recording times. In addition, the RMSSD time domain is a robust value for interpreting the parasympathetic response of a subject, which makes it an important parameter to determine the variation of autonomic control in a human being. The citations exposed in the text to justify their use support the mentioned background. |
|
(Table 1) Please, write the unit of each variable |
The units in table 1 were added as suggested. |
|
(line: 298) according to the inclusion criteria, all participants were free from cardiovascular diseaes! |
The wording used in the conclusion was modified to better reflect the content and methodology of the manuscript. |
Round 2
Reviewer 1 Report
Congratulations. The manuscript was improved, and I recommend its acceptance.
Reviewer 3 Report
Well done! The authors addressed all the comments. Just a minor piece of advice, avoid long sentences and keep the writing simpler.